# Hairy Fluorescent Nanospheres Based on Polyelectrolyte Brush for Highly Sensitive Determination of Cu(II)

**DOI:** 10.3390/polym12030577

**Published:** 2020-03-05

**Authors:** Qiaoling Wang, Kaimin Chen, Yi Qu, Kai Li, Ying Zhang, Enyu Fu

**Affiliations:** 1College of Chemistry and Chemical Engineering, Shanghai University of Engineering Science, Shanghai 201620, China; woodsues@outlook.com (Q.W.); 18301939658@163.com (K.L.); fuenyu1234@163.com (E.F.); 2School of Chemical Engineering, East China University of Science and Technology, Shanghai 200237, China; zy12fearless@163.com

**Keywords:** fluorescent nanosphere, 5-aminofluorescein, copper ion detector, coupling

## Abstract

Currently, it is an ongoing challenge to develop fluorescent nanosphere detectors that are uniform, non-toxic, stable and bearing a large number of functional groups on the surface for further applications in a variety of fields. Here, we have synthesized hairy nanospheres (HNs) with different particle sizes and a content range of carboxyl groups from 4 mmol/g to 9 mmol/g. Based on this, hairy fluorescent nanospheres (HFNs) were prepared by the traditional coupling method (TCM) or adsorption-induced coupling method (ACM). By comparison, it was found that high brightness HFNs are fabricated based on HNs with poly (acrylic acid) brushes on the surface via ACM. The fluorescence intensity of hairy fluorescent nanospheres could be controlled by tuning the content of 5-aminofluorescein (5-AF) or the carboxyl groups of HNs easily. The carboxyl content of the HFNs could be as high as 8 mmol/g for further applications. The obtained HFNs are used for the detection of heavy metal ions in environmental pollution. Among various other metal ions, the response to Cu (II) is more obvious. We demonstrated that HFNs can serve as a selective probe and for the separation and determination of Cu(II) ions with a linear range of 0–0.5 μM and a low detection limit of 64 nM.

## 1. Introduction

Heavy metal ion determination and removal are becoming more and more important. Copper (Cu(II)), a crucial cofactor for multiple enzymes, plays an critical role in bone formation together with certain proteins [1,2], but it is considered a pollutant when the concentration is over a critical level in the environment. It is easy for Cu(II) to form a complex that can change the molecular structure of a protein and break hydrogen bonds. As a result, biological reactions and the human body, including the kidneys, liver skin, bones and teeth, are affected by Cu(II) [3,4]. The allowable limit of Cu(II) ions in potable water is 2 mg/L (World Health Organization, WHO), but the maximum permissible level restricted by the United States of Environmental Protection Agency (USEPA) is only 1.3 mg/L [5,6,7]. Thus, it is essential to develop effective technologies to detect Cu(II) from waste water before discharging it into the environment to safely protect the community health [8]. 

Over the last decade, many analytical approaches for Cu(II) ions quantification have been noted according to the detection methods, such as atomic absorption spectroscopy (AAS) [9], colorimetric, fluorescent methods [10], strip and magnetic resonance imaging (MRI) sensor ensemble materials [11]. The fluorescent methods, especially fluorescent nanosphere probes, are promising candidates due to the excellent selectivity, easy operation and fast analysis of the fluorescence techniques.

Fluorescent nanospheres (FNs) that can excite fluorescence under the stimulation of external energy have become of interest in many fields, particularly in ion detection [12,13,14,15]. Recently, many groups have been devoted to preparing FNs by different methods, such as encapsulating fluorescent molecules or quantum dots into the matrix spheres [16,17,18] and placing nanoparticles on the surface of matrix spheres [19,20]. In consideration of differentiated application environments, FNs are fabricated based on different matrices, including polystyrene [21], polymethyl methacrylate [22], silica [23], polylactic acid [24], melamine formaldehyde [25], sodium alginate [26], conjugated microporous polymers [27] and nanocellulose [28]. Each type of matrix spheres and preparation methods has advantages and disadvantages, with respect to the cost-effectiveness, properties and stability of FNs, as shown in Table 1. Seeking effective preparation methods based on a given matrix with special properties to meet the requirements of desired properties is of great significance.

Hairy nanospheres (HNs) based on polymer brush structure, typically spherical poly (acrylic acid) brushes [29,30], have attracted much interest by virtue of their properties, including high stability in various hydrophilic systems and abundant functional carboxyl groups. HNs with ultrahigh surface carboxyl groups can be coupled to fluorophores directly without introducing reactive groups on the surface, and residual carboxyl groups are convenient to use in other preparation processes or in the application of FNs. Therefore, HNs with rich carboxyl groups [31] and special three-dimensional structures can be considered as promising host materials for FNs and have broad application prospects in the field of ion detection.

In this paper, we report an effective method for fabricating hairy fluorescent nanospheres (HFNs) using hairy nanospheres (HNs). The HFNs were well characterized and the fluorescence intensity was controlled and optimized. The method presented here can be considered as a general method to other fluorescent molecules with functional groups. The extra carboxyl groups of HFNs can be used in further applications like enzyme immobilization, antibody coupling etc. Here, HFNs with abundant extra carboxyl groups could be used as the detector for Cu(II).

## 2. Materials and Methods

### 2.1. Materials

N-hydroxy-succinimide (NHS), potassium persulfate (KPS), hydrochloric acid (HCl), sodium hydroxide (NaOH), sodium dodecyl sulfate (SDS), 1-ethyl-3-(3-dimethylaminopropyl) carbodiimide hydrochloride (EDC) and 2-(N-morpholino) ethanesulfonic acid (MES) were bought from Adamas-beta (Shanghai, China). 5-Aminofluorescein (5-AF) was obtained from Aladdin (Shanghai, China). Sodium hydrogen phosphate and sodium dihydrogen phosphate were obtained from sinopharm (Shanghai, China).

Styrene (St) and acrylic acid (AA) were purchased from Aladdin (Shanghai, China) and reserved to use after distillation under reduced pressure. Phosphate buffer (PB, pH = 7.2) and 2-(N-morpholino) ethanesulfonic acid buffer (MES, pH = 5.5) were formulated before the coupling experiments.

### 2.2. Synthesis of Hairy Nanospheres (HNs)

Narrowly distributed hairy nanospheres (HNs) were prepared by the combination of conventional emulsion polymerization and photoemulsion polymerization according to our previous work [32,33,34]. First, a photoinitiator was generated on the core under the conditions of avoiding light. The system was purified by ultrafiltration using deionized water until the conductivity was no longer changed. Then the calculated doses of the monomer AA were added into a photoreactor containing a purified polystyrene core with a photoinitiator, and the entire system was illuminated by UV−visible radiation with constant stirring under a dark environment. After completion of the polymerization, the hairy nanospheres (HNs) were purified by dialysis using pure water until the conductivity no longer changed.

### 2.3. Synthesis of HFNs by Traditional Coupling Method

Typically, for the traditional coupling methods (TCM), 10 mg of hairy nanospheres (HNs) were added into 10 mL buffer (MES, Ph = 5.5), then 80 mg NHS and 120 mg EDC were dispersed into the buffer. The mixture was incubated for 15 min under shaking at room temperature. 5-AF, ranging from 0.5 wt.% to 4 wt.%, was dispersed into 10 mL PBS buffer (pH = 7.2), then HNs after activation of carboxyl were quickly added into the solution containing 5-AF under sonication. The solution was allowed to react for 10 h under gentle shaking at room temperature. The HFNs were purified by dialysis using pure water until the conductivity was no longer changed. 

### 2.4. Synthesis of HFNs by Adsorption-Induced Coupling Method

For the adsorption-induced coupling methods (ACM), the 10 mg HNs were dissolved in buffer (MES, pH = 5.5) and 5-AF, ranging from 0.5 wt.% to 16 wt.%, was added. The mixture was subjected to physical adsorption for 2 h at room temperature and then coupled with 80 mg EDC on a shaker for 10 h to obtain narrowly dispersed hairy fluorescent nanospheres (HFNs). After the coupling was completed, the HFNs were purified by dialysis using pure water until the conductivity was no longer changed. 

### 2.5. Characterization

The hydrodynamic diameter of particles was measured by dynamic light scattering (DLS NANO-S90). Intensity-average size and polydispersity index (PDI) were obtained for analysis.

Fluorescence images of HFNs was obtain by laser confocal microscope calorimetry (LEICA TCS SP8, excitation with 488 nm excitation, emission range from 500–550 nm).

Morphology of particles were characterized by scanning electronic microscopy (SEM), the voltage of the instrument is about 5 kV.

The carboxyl content of the polyelectrolyte brush was calculated by conductivity instrument (METTLER-T50). The titrant is NaOH and its concentration is 0.1 mol/L.

The fluorescence intensity of all HFNs were measured by the constant wavelength synchronous fluorescence method (Spectrofluorometer FS5). In our study, we chose a constant-wavelength (Δλ) equal to the Stokes shift (the excitation wavelength is 490 nm, emission wavelength is 517 nm, Δλ = λem - λex = 27 nm).

### 2.6. Fluorescent Assays

The probability of HFNs as a fluorescent detector for metal ions was investigated at a normal temperature. Some metal ions solutions, including Cr(III), Cu(II), Cd(II), Ca(II), Mg(II), Pb(II), Ni(II) and Ag(I),were mixed with FHNs respectively with same concentrations (10 µM of metal ions and 2 × 10^10^ particles mL^−1^ of FHNs at most). 

The quantity of HFNs in aqueous solution can be calculated from the formula (1) [35].
The quantity of HFNs per mL = 3*ω*/(4*πR*^3^*ρ*)(1)
where, *ω* is the grams of polystyrene core in the HFNs added per milliliter, *R* is the average radius of the polystyrene core, 35.5 nm, and *ρ* is the density of the polystyrene core, 1.05 g/cm^3^. In order to detect Cu(II) accurately, metal ions with different concentrations were mixed with HFNs at room temperature (the concentration of HFNs was kept at 2 × 10^10^ particles mL^−1^).

After shaking for 3 min at normal temperature, fluorescent signals were recorded by using the synchronous fluorescence method (Spectrofluorometer FS5). 

## 3. Results and Discussion

### 3.1. Synthesis and Characterization

As shown in the Figure 1a, HFNs with rich fluorescent molecules were obtained in this work. Morphology of corresponding nanospheres are shown in the Figure 1b, the obtained HFNs have a narrow distribution and uniform particle size. In fact, SEM could not reflect the true size of the polymer brush layer in the aqueous solution, because the dry state of the polymer brush plus the fluorescent molecules can hardly be seen in SEM; however, the DLS could provide a hydrodynamic size of the nanospheres and was depicted in the illustration inserted in the SEM of Figure 1b. From the Fourier transform infrared spectrometer (Appendix A) and confocal laser scanning microscope (CLSM), the changes in the infrared absorption peaks and the appearance of green fluorescence could be clearly observed, indicating the successful coupling of fluorescent molecules in the polymer brush structure. 

### 3.2. Effect of Coupling Method

In this work, hairy nanospheres and polystyrene-poly (acrylic acid) (PS-PAA) were selected as a carrier. The core of HNs used here is 157 nm, and the thickness of the PAA chain is 57 nm. The amino of 5-AF and carboxyl of HNs can be fixed together by typical NHS/EDC coupling. However, the process may greatly sacrifice the potential of the PAA brushes due to the special three-dimensional structure of the HNs. Therefore, two methods for preparing HFNs were designed, i.e., the traditional coupling method (TCM) and adsorption-induced coupling method (ACM). 

The proposed mechanism for TCM is described in Scheme 1. After the addition of EDC and NHS (Scheme 1a), the carboxyl groups of the HNs were transferred into an ester intermediate immediately under the activation of EDC, and in the next step, the ester intermediate was reacted with the amino group on the 5-AF to form a stable amide bond [36,37,38,39,40]. The 5-AF was first coupled to the binding site of the outer layer of the HNs, which reduced the electrostatic property of HNs and the extra 5-AF in the solution was difficult to diffuse into the inner layer of PAA with the electrostatic driving force. On the other hand, the steric hindrance produced by 5-AF coupled to the outer layer of HNs could also retard the diffusion of 5-AF and dramatically affect the coupling process. In a word, 5-AF could only form a monolayer coupling at the out layer of FNs, and residual 5-AF would be removed during the dialysis process and could not contribute to the fluorescence intensity even if the original dosage of 5-AF is at a high level.

However, in the adsorption-induced coupling method (ACM), both the HNs and the 5-AF were dispersed in the MES buffer; therefore, it was easier for 5-AF to enter the inside of the HNs and carry on electrostatic adsorption owing to the unique electrostatic adsorption of the polyelectrolyte brushes [41,42,43]. The HNs with plentiful carboxyl groups were considered specifically advisable for this method, as the steric hindrance was no longer dominant in this process. Physical adsorption [44,45,46] could enable the 5-AF to enter the inside of the hairy nanospheres easily, and their positions in the polyelectrolyte could be rearranged. Therefore, a large amount of 5-AF could be embedded in the PAA structure by the electrostatic driving force. Once the EDC was added in the next step, the carboxyl groups at these unfavourable sites were activated immediately to form a strong amide bond due to high activity of EDC in the buffer at pH = 5.5. 

To confirm the speculation, the two coupling methods were carried out and compared. As a start, the traditional coupling method (TCM) was employed to introduce 5-AF into the monodisperse HNs bearing poly(acrylic acid) (PAA) shell with thickness of 57 nm.

The concentration of the 5-AF was firstly set to 0.5 wt.%, which is a typical dye content in fluorescent nanospheres obtained by other methods [47,48], as shown in Figure 2a. The 5-AF content was then increased up to 4 wt.% to make full use of the abundant carboxyl groups. The fluorescence intensity of the hairy fluorescent nanospheres (HFNs) obtained by the TCM method increases slightly as the 5-AF dosage increases. Thus, the 5-AF could not contribute to the fluorescence intensity even the original dosage is at a high level. It is difficult for the TCM method, in which NHS/EDC was used as the coupling agent, to take full advantage of the ultrahigh functional groups on the HN surface.

We then changed to an alternative coupling process [46] (adsorption-induced coupling method, ACM) to improve the coupling efficiency. Then HFNs with 4 wt.% 5-AF were obtained and there is a remarkable increase in the fluorescence intensity when compared to TCM (Figure 2b). The selected PAA thickness is 50 nm in this study, and the fluorescence intensity of HFNs synthesized by the adsorption-induced coupling method (ACM) is 200% of the traditional coupling method. Table 2 shows the diameters of the HFNs obtained by different methods. When the same content of 5-AF is added, the particle size of the HFNs obtained by the TCM is smaller than before, and the PDI distribution becomes wider. The diameter of the HFNs obtained by ACM is larger than by TCM, and the PDI distribution is basically unchanged. This is in line with our previous theory: 5-AF could be embedded in the PAA structure easily by the electrostatic driving force for ACM. Uniformly developed and stable ultrabright HFNs are obtained in the work. Then, ACM was selected as the suitable coupling method for further study of optimization of the fluorescence properties of HFNs with various PAA thicknesses in different HNs or various 5-AF dosages in the coupling process. 

### 3.3. Effect of the Carboxyl Content of HNs

Fluorescein is connected to the carboxyl in the HNs. Therefore, the fluorescein immobilization amount is closely related to the carboxyl content, so the effect of carboxyl content on fluorescence intensity was explored in the further study. HNs with different PAA thickness were used and their characteristic parameters are shown in Table 3. the polystyrene core was 77 nm with a narrow PDI of 0.052 (the DLS profile is shown in Figure 3a) and the PAA thickness ranged from 34 nm to 139 nm. The conductivity change of the HNs was then measured by conductometric titration. The titration curve is shown in Figure 3b.

The carboxyl content of the HNs is calculated according to formula (2).
Carboxyl content (mmol/g) = (*V*_2_ − *V*_1_)/*m C*(2)
where *C* is the concentration of the titrant (HCl), *V*_2_ and *V*_1_ are the volumes consumed by sodium hydroxide, which are the turning points at the low conductivity after the addition of the NaOH and *m* is the amount of added HNs (g).

The conductometric titration curves of HNs with various PAA thickness was shown in Figure 3b. It was calculated that the carboxyl content ranges from 4 mmol/g to 9 mmol/g. A fixed dosage of 5-AF (4 wt.%) was used to couple with HNs with various PAA thickness in the ACM method. As shown in Figure 4a, with a fixed 5-AF dosage, the fluorescence intensity of HFN increases as the carboxyl content of the HNs increase. The reason for the increase is that thickened HNs reduce the number of fluorescent molecules per unit volume, meaning that the distance between the 5-AF molecules is larger after the process of physical adsorption, avoiding the occurrence of possible self-quenching [49]. At the same time, the increase in the thickness of HNs increases the capacity for fluorescent molecules on the HNs. Therefore, for HNs with the same core size, the thicker the PAA, the higher the carboxyl content and the higher the fluorescence intensity of the obtained HFNs, as shown in Figure 4b. HNs with the largest particle size of 355 nm were then selected as the carrier of 5-AF to understand the relationship between the 5-AF dosage and the fluorescence intensity of HFNs.

### 3.4. Effect of Dosage of 5-AF

The fluorescence intensity of the HFNs also depends on the dye content. A high content of 5-AF coupled to the carboxyl groups may lead to fluorescence quenching, which is adverse to obtaining a cost-effective use of 5-AF. We studied the relationship between the 5-AF dosage and fluorescence intensity of the HFNs. The core of HNs used here is 77 nm and the thickness of PAA chain is 139 nm. As shown in Figure 5a, continuous fluorescence enhancement and a red shift of the peak from 482 to 486 nm were observed upon the addition of 5-AF from 0.5 wt.% to 8 wt.%. 

As shown in Figure 5b, when the dosage of 5-AF was less than 4 wt.%, a quasi-linear relationship between the reacted carboxyl content and the dosage of 5-AF was found in this study. As we continued to increase the dosage of 5-AF, the reacted carboxyl content slightly increased. When the dosage of 5-AF was less than 8 wt.%, the fluorescence intensity of HFNs increased with the increase of 5-AF dosage. When the dosage of 5-AF was 16 wt.%, the fluorescence intensity suddenly decreased at 3 mmol/g of reacted carboxyl content, which is not consistent with the consumed carboxyl groups. With the increase of 5-AF, it was found that the reacted carboxyl content of HNs gradually increased and reached a plateau near 3 mmol/g. When the dosage of 5-AF was doubled (from 8 wt.% to 16 wt.%), the consumption of carboxyl groups did not increase significantly, i.e., the majority of the added 5-AF did not react with the carboxyl groups. 

The saturated reaction amount of carboxyl groups is 3 mmol/g, which is near one-third of the total carboxyl groups. When the 5-AF dosage continues to increase from 8 wt.% to 16 wt.%, the fluorescence intensity sharply decreased, which is most likely as the result of the fluorescence aggregation quenching effect. The spacing between the fluorescent molecules reaches the critical point of fluorescence quenching due to the further increase of 5-AF. When the PAA thickness is 139 nm with the 8 wt.% 5-AF, the highest fluorescence brightness HFNs can be obtained.

We then calculated that up to two-thirds of the carboxyl groups were still free after adequate coupling of 5-AF, which could be used as the functional groups in applications including ion detection and removal.

### 3.5. Fluorescence-Based Detection of Cu(II) Ions

In this work, HFNs (HN5 with 8 wt.% 5-AF) were selected as a detector to detect Cu(II). The fluorescence intensity of HFNs versus the concentration of Cu(II) (original fluorescence spectra is shown in Figure 6a). As we continued to increase the concentration of copper ions, the fluorescence intensity of HFNs decreased gradually. As shown in Figure 6b, the quenching efficiency (*F*_0_ − *F*)/*F*_0_ was found to be linear with respect to the concentration of Cu (II) (0–0.5 μM, *R*^2^ = 0.997). Therefore, HFNs can be used as a sensitive fluorescence detector for detecting the concentration of Cu(II). There is a good linear relationship between the fluorescence enhancement of Cu(II) loaded HFNs (HFNs@Cu(II)) and the concentration of S(-II) (0–0.5μM, *R*^2^ = 0.970). The recovery of HFNs from HFNs@Cu(II) could be realized by further adding sulfide anion S(-II) [50] into the HFNs@Cu(II) system. (Figure 6c,d). Apart from the sensitivity, it is also necessary to analysis the probability of reusability of the fluorescence detector [49] at a cost-effective point of view. Six sequential recovery/quenching cycles were successfully performed to remove Cu(II) from the HFNs by using S(-II) (Figure 6e). The results showed that the HFN still remained 60% of the original fluorescence intensity after six cycles. 

The detection performance of HFNs was good compared to other reported results concerning the fluorescence sensors for the detecting of Cu(II) or S(-II) as shown in Table 4. Similar to most fluorescence detectors that have been reported in the literature, it is selective for target ions during the detection of metal ions. The HFNs provide a lower detection limit. They have a significant advantage in terms of both recovery and reusability. These data indicate that HFNs have excellent performance in detecting Cu(II), and the detection limit of copper in aqueous solution is 64 nm, which is much lower than the standard set by WHO. HFNs@Cu(II) could be recovered by S(-II) for recyclable use.

Various metal ions [51,52] were selected to study their sensitivity to HFNs and the results are shown in Figure 6f. We found that the intensity of HFNs is very sensitive to Cu(II) compared with the other metal ions, which is also reported elsewhere [53,54,55]. The fluorescence selectivity was investigated to understand the behavior of HFNs and HFNs dispersed in different cation Ag(I), Cd(II), Cu(II), Ca(II), Cr(III), Mg(II), Ni(II) and Pb(II) solutions. While most cations caused slight quenching in the intensity of HFNs, dramatic fluorescence quenching can be observed after the addition of Cu (II) (Figure 6f). The results also suggested the selectivity of the HFNs towards Cu(II) ions. The reason that Cu(II) ions were sensitively preferred by HFNs is that the ions in the ligand nitrogen are more attractive to Cu(II), resulting in a more stable complex in Cu(II) ion adsorption [56], which may give a reasonable mechanism of Cu(II) ion and the HFNs active site at the specific conditions.

## 4. Conclusions

In summary, a new adsorption-induced coupling method (ACM) for the preparation of fluorescent nanospheres has been developed. The ACM makes full use of the abundant carboxyl groups, which means that it is efficient, simple and easy to control the fluorescence intensity of HFNs. When the PAA thickness is 50 nm, the fluorescence intensity of HFNs obtained by the adsorption-induced coupling method (ACM) is 200% of the traditional coupling method (TCM). The thicker the PAA, the higher the carboxyl content, and the higher fluorescence intensity at the same 5-AF dosage. For HNs with the same PAA thickness, 5-AF dosage could be higher, up to 8 wt.%, without obvious fluorescence quenching, which is almost one magnitude higher than that produced by other routes. HFNs with abundant extra carboxyl groups could be used as detectors for Cu(II). Extra carboxyl groups on the surface of HFNs can also be used for further applications, and the method proposed here can be considered as a general method for other fluorescent molecules with functional groups. By adjusting the 5-AF content and the PAA thickness, fluorescent-coded beads can be formed for biological analysis. At the same time, HFNs provide useful guidance for understanding the principles of fluorescence tuning and designing novel fluorescent nanospheres in the fields of labeling, detection and biosensing.

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
