# Peer review of "Hairy Fluorescent Nanospheres Based on Polyelectrolyte Brush for Highly Sensitive Determination of Cu(II)"

_polymers, 2020, doi:10.3390/polym12030577_

Round 1

Reviewer 1 Report

The current paper of Qu and colleagues on the synthesis of hairy fluorescent nanospheres based on 2 polyelectrolyte brush for highly sensitive determination of Cu(II) presents reasonably valuable scientific results, however the following important comments as a major revision should be considered prior acceptance of the manuscript:
1. First of all in order to further improve the manuscript, initially I would like to suggest to improve the English; explicitly the text should be checked by a native English speaker before further submission.
2. Critically, it is hard to follow the storyline of the manuscript; indeed the introduction is not well-structured, it is failing to be concise and clarify the motivation for the work presented, therefore the introduction should be modified accordingly.
3. The authors claim that the HFNs with rich fluorescent molecules obtained in this work have narrow distribution and uniform particle size, but any particular value is not given. Please provide exact data about this information.
4. The reasoning to choose the traditional coupling method (TCM) and adsorption-induced coupling method (ACM) as methods of choice to prepare the HFNs is not well structured, please be more specific.
5. The Donnan effect is not clearly discussed within the following sentence, please explain it concisely: However, in the adsorption-induced coupling method (ACM), both the HNs and the 5-AF were dispersed in MES buffer, it was easier for 5-AF to enter the inside of the HNs and carry on electrostatic adsorption owing to the unique "Donnan effect" of polyelectrolyte brushes[40-42].
6. The authors claim that it is difficult for TCM method, in which NHS/EDC was used as the coupling agent, to take full advantage of the ultrahigh functional groups on HNs surface, hence was it considered any other method for the couplings, for instance CDI mediated coupling
7. What do the authors intend to mean with “quasi-linear” in the following sentence:
It was calculated that the carboxyl content ranges from 4 mmol/g to 9 mmol/g, which is quasi-linear to the hydrodynamic diameter of HNs.
8. Could the authors justify with any analysis that the self-quenching is completely avoided for HNs with the largest particle size of 355 nm?
9. Could the authors give more specified future applications based on HFNs?

Reviewer 2 Report

The manuscript is reporting the simple and easy preparation of hairy fluorescent nanospheres (HFNs) using adsorption-induced coupling method (ACM) which is more efficient in terms of higher fluorophore (5-AF) conjugation than that of the traditional coupling method (TCM). The nanospheres are also applied for fluorescence quenching induced detection of Cu(II) to form the Cu(II) complex which was further used for emission enhanced sensing of sulfide. The reporting of the corresponding material synthesis and its potential applications are novel, thus the manuscript should be accepted for publication in this journal. Unfortunately, manuscript preparation needs proper revision prior to publication. So, it is recommended to revise the manuscript by addressing the below mentioned comments.

Specific comments:                   

The representation of sulfide anion as “S(II)” should be verified with proper references. Otherwise, please use the general representation of sulfide as “S2“. Page 3, in the section of “Synthesis of hairy fluorescent nanospheres (HFNs)”, the mentioning of “10 mg fluorescent nanospheres (HNs)” should be "10 mg hairy nanospheres (HNs)". Page 3, in the “Characterization” section, the first sentence should be completed. It is also mentioned that “In our study, choose Δλ equal to the Stokes shift (Δλ= 27 nm)”. What does it mean? In the Figure 1 caption, mention the details of microscopic imaging (CLSM). In Figure 2, what is the reason of reduced fluorescence intensity in case of 4 wt% 5-AF used in adsorption-induced coupling method (ACM) compared to that of the 2 wt% 5-AF used? What is the size of the HNFs used here? In Figure 4 caption, please mention the source (preparation methods) of HFNs. What is NH1-5 in Figure 4a? Should it be HFN1-5, the corresponding AF-modified HN1-5? Also please mention the significance of circular and triangular spots respectively in Figure 4b. To prepare the HFN by ACM method for Figure 4, the fixed dosage of 5-AF was 4 wt%. But from Figure 2, it is observed that 4 wt% use of 5-AF shows reduced fluorescence intensity compared to that of 2 wt%. So, the authors are recommended to reproduce this data using 2 wt% of 5-AF. Please mention the size of the HFNs used for the experiments and the experimental conditions (solvent, excitation wavelength, etc) in each figure captions. What are the concentrations of the various metal ions used for measuring selectivity in Figure 6f? What about the fluorescence chnages in case of addition of Zn(II) to the HFNs? Zn(II), Fe(II) and Fe(III)should be included in the selectivity measurement graph in Figure 6f. See the related competing analytes in a recent literature on fluorescence detection of Cu(II): Sensors and Actuators B: Chemical, 2019, 279, 204-212. In Table 2, mention the indication of superscript "1" for MES buffer. Also mention the importance of use of acidic buffer (MES, pH = 5.5) in the preparation of HFNs.

Reviewer 3 Report

Please rewrite the Abstract in a more specific way.  Correct also the sentence "And the fluorescent" as: "The fluorescent..."

In Introduction section, please reformulate the first paragraph in a concise manner, without undesired repetitions.

Please distribute the references [8-10] to the specific detection.

Rewrite in a clear sentence the text between lines 42-43. Introduce a Table to present the performances/ disadvantages of the materials prepared in references 20-27. The text from lines 51-55 should be removed it is too general.

The text between 58-60 lacks in clarity and is too general. Please explain in detail " without further modification" .........

The goals are not appropriately established. Is this a method for Cu(II) detection, or for S detection? Describe the material on which is based the HFNs. Please do not use And at the beginning of the sentence. remove all the "And".

Section 2.2. Please explain if the light is avoided or the system is illuminated by UV-vis radiation??? here is a contradiction in preparation of HNs. Complete the data with quantities (mole), molar ratios. What is the appropriate amount of 5-AF? The data are not complete in order to assess the results.

Organize this Syntheses part in three distinct sections.

All the text in paragraph 148-152 is a repetition.

The terms are imprecise; is it fluoresceine or 5-fluoresceinamine ? The text produces confusion. What is the fixed 5-AF dosage?

A table should  be prepared to present all the prepared samples and their composition and other characteristics. The reaction of functionalization with 5-AF should be monitored carefully.

The prepared materials are not discriminated in function of composition, their thickness... The method for detection is not appropriately described.

What are the concentrations for each studied metal?

The paper has to be rewritten to provide clarity. Systematic approach experiments and results is needed.

Round 2

Reviewer 1 Report

Please accept the manuscript in its current form

Author Response

Many thanks for your comments.

Reviewer 2 Report

The revised manuscript is acceptable for publication in this journal.

Author Response

Many thanks for your comments.

Reviewer 3 Report

The authors provide an improved manuscript, nevertheless many aspects need correction.

  1. HFNs were never introduced as an abbreviation and it is difficult to discriminate between HNs ( that are wrongly introduced in ABSTRACT=Hairy fluorescent nanospheres) and HFNs as they are presented in the text. Please carefully corect these confusing aspects.
  2. Introduction: When presenting the purpose of the paper, please be precise and reformulate the whole text between lines 84-87. Please do not begin sentences with 'And".

3. In Table 1 --The DIAMETER of HNs: Where is the Diameter?? The author cannot name a Table with Diameter and not to provide that data.

4. The Synthesis is not provided with the required information. Section 2.3 What is the Quantity of 5-AF? The same aspect for Section 2.4.

5 Section 3.5 -Title== please correct : was selected as sensitive material to detect Cu(II)

6. The whole paragraph from lines 365 to 374 has to be revritten. It has no meaning . It is not enough to cut Sulfur from the text. Please reformulate.

7. Figure 6 e has no meaning in the actual context.

8. The same aspect mentioned at point 5 for the paragraph  from 385 to 405.

Round 3

Reviewer 3 Report

The manuscript entitled: Hairy fluorescent nanospheres based on polyelectrolyte brush for highly sensitive determination of Cu(II), was highly impoved,

but I still have a great concern about methodology.

So, The Sub -chapter 2.3 and 2.4 have to be re-written providing all the data.

It cannot be accepted something not clearly written, such as:: then the appropriate NHS and EDC were dispersed into the buffer.

Besides the amounts of AF (0.5 wt% reported to XXXXX), have to be reported to another substance.
